# Design of a Dynamic Secondary Mirror Truss Adjustment Mechanism for Large Aperture Telescopes

**Baowei Lu** [1,2], **Fuguo Wang** [1,*] **and Benlei Zhang** [1,2]

1   Changchun Institute of Optics, Fine Mechanics and Physics, Chinese Academy of Sciences, Changchun 130033, China
2   University of Chinese Academy of Sciences, Beijing 100049, China
*   Correspondence: wfg109@163.com

**Abstract:** This study focuses on the dynamic secondary mirror truss adjustment mechanism for large aperture telescopes. The relative positions between the primary and secondary mirrors have stringent requirements for large aperture optical telescopes, with an aperture of more than 2 m. Due to the large mass of the primary mirror, the secondary mirror system is designed as an adjustable mechanism with multiple degrees of freedom, dramatically impacting telescope imaging. The kinematic modeling is followed by a detailed description of the designed adjustment mechanism, and then a static and modal analysis of the designed mechanism is performed. Subsequently, a kinematic performance test of the experimental prototype is conducted. The developed mechanism can travel up to $\pm 5$ mm in the z-direction with an accuracy of 16 μm and deflection of $\pm 0.574°$ in the xy-direction. It provides accuracy better than 6.4 arcseconds.

**Keywords:** large aperture optical telescope; secondary mirror system; dynamic secondary mirror truss adjustment mechanism

## 1. Introduction

The secondary mirror must maintain good face accuracy against ambient temperature changes, manufacturing limitations, installation, gravity, heat transfer, and material properties. These factors can cause a relative attitude shift between the primary and secondary mirrors, leading to an optical axis tilt, thus affecting the system's pointing accuracy and imaging quality. Therefore, the relative positions between the primary and secondary mirrors must be corrected. Due to the large mass, it is difficult to move the primary mirror. In contrast, due to its small size, the secondary mirror can be adjusted in multiple degrees of freedom (DOFs). As an essential part of the telescope system [1], the secondary mirror supports and changes its position. It must maintain a close range of position and attitude, thus ensuring good imaging quality of the telescope system [2]. The adjustment mechanism includes piston and tip/tilt, which eliminates third-order coma aberrations, and enables the elimination of third-order comet aberrations caused by deviations in the position of the primary and secondary mirrors. It is essential to consider the chopping of the rapid oscillation of the secondary mirror in IR modulation technology. Therefore, the adjustment mechanism of the secondary mirror must be designed as a system with at least three DOFs. The BLAST-TNG telescope uses three linear actuators to adjust the three DOFs of the secondary mirror piston/tip/tilt [3,4]. The William Herschel Telescope uses four Focus Translation Units to correct the primary focus for the three DOFs of the piston tip/tilt [5,6]. The Stewart stage is employed in the secondary mirror adjustment mechanism (SMAM) of large aperture telescopes, such as VST telescopes [7–9] and LSST telescopes [10,11]. It offers many advantages, such as high stiffness, six degrees of freedom of adjustment, and non-accumulation of position errors [12]. In addition to five DOFs of adjustment, the SOFIA sub-mirror has a fast chopper with frequencies up to 20 Hz [13,14].

This paper deals with the design of the secondary mirror adjustment in the context of a large vehicle-mounted telescope. It is an optical telescope with an aperture greater than 2 m and is used for applications with significant aberrations. First, the kinematic modeling of the mechanism is conducted. The designed dynamic secondary mirror truss adjustment mechanism is analyzed through static and modal analyses in finite element analysis software. Then, kinematic performance tests on the test prototype are performed to verify the movement accuracy of the designed mechanism requirements. The presented study provides a guideline for SMAMs and large aperture telescopes.

## 2. Kinematic Modeling of the Mechanism

### 2.1. Kinematical Modeling

The adjusting mechanism is a parallel mechanism consisting of a fixed platform, a moving platform, and four struts connecting the moving platform and the fixed platform, each strut containing a moving sub (P), a universal sub (U), and a rotating sub (R). The sketch of this adjustment mechanism is shown in Figure 1. In the figure, $A_1$, $A_2$, $A_3$, and $A_4$ are the intersection points of the four moving subs and the fixed platform. $B_1$, $B_2$, $B_3$, and $B_4$ are located on the R sub at the intersection point of the fixed platform and the four branch chains. $C_1$, $C_2$, $C_3$, and $C_4$ are at the center of the four U subs. The quadrilateral $A_1A_2A_3A_4$ and $B_1B_2B_3B_4$ are both squares, and O and O' are the center points of the squares $A_1A_2A_3A_4$ and $B_1B_2B_3B_4$, respectively. A fixed coordinate system O-xyz is set up in the $A_1A_2A_3A_4$ plane with the point O at the center, where the x- and y-axes point to a branch chain, and the z-axis points vertically upwards. A fixed coordinate system O'-x'y'z' is established in the $B_1B_2B_3B_4$ plane with the point O' at the center, the x'- and y'-axes pointing to a branch chain and the z'-axis pointing vertically upwards.

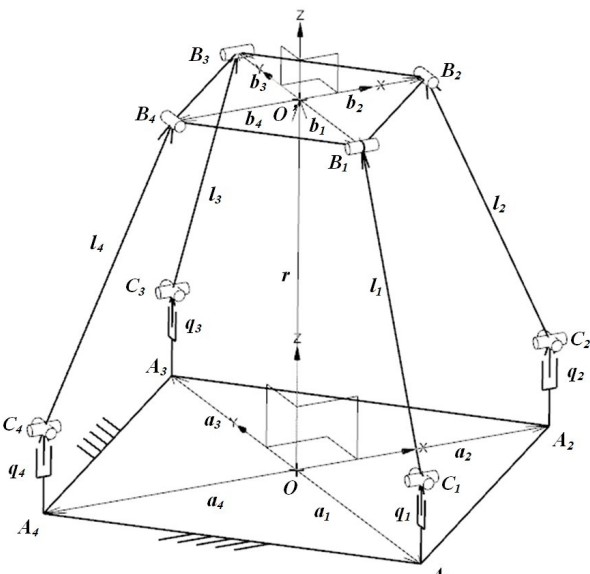

**Figure 1.** Sketch of the adjustment mechanism.

A fixed angle transformation is applied to describe the position of the dynamic coordinate system O'-x'y'z' about the fixed coordinate system O-xyz. Where $\psi$ is the angle of rotation around the x-axis, $\theta$ around the y-axis, and $\varphi$ around the z-axis, these angles develop the attitude matrix as follows:

$$\mathbf{R} = \mathbf{R}_z(\varphi)\mathbf{R}_y(\theta)\mathbf{R}_x(\psi)$$
$$= \begin{bmatrix} c\varphi c\theta & c\varphi s\theta s\psi - s\varphi c\psi & c\varphi s\theta c\psi + s\varphi s\psi \\ s\varphi s\theta & s\varphi s\theta s\psi + c\varphi c\psi & s\varphi s\theta c\psi - c\varphi s\psi \\ -s\theta & c\theta s\psi & c\theta c\psi \end{bmatrix},$$

(1)

where "$s$" and "$c$" denote sine and cosine, respectively.

### 2.2. Analysis of the Model Degrees of Freedom

The mechanism consists of four identical PUR-branched chains connected to the fixed and moving platforms. Figure 1 shows each branched chain. These are equivalent to three single DOF rotations and one moving sub, as shown in Figure 2. The intersection of $\$_{i2}$ and $\$_{i3}$ with the origin establishes the branched chain coordinate system O-$x_i y_i z_i$; then, the motion spiral of the moving sub of PUR-branched chain "$i$" system is defined as follows:

$$
\begin{aligned}
\$_{i1} &= \begin{pmatrix} 0 & 0 & 0; & 0 & 0 & f_1 \end{pmatrix} \\
\$_{i2} &= \begin{pmatrix} 1 & 0 & 0; & 0 & 0 & f_2 \end{pmatrix} \\
\$_{i3} &= \begin{pmatrix} 0 & 1 & 0; & 0 & 0 & f_3 \end{pmatrix} \\
\$_{i4} &= \begin{pmatrix} 0 & 1 & 0; & d_4 & 0 & f_4 \end{pmatrix}
\end{aligned}
\tag{2}
$$

where $f_1, f_2, f_3, f_4$, and $d_4$ are all non-zero real numbers.

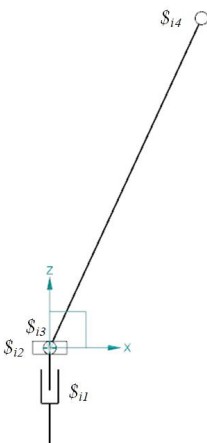

**Figure 2.** Four PUR-branched chains.

Using the principle that the reciprocal product of spirals is zero for the kinematic spiral system of a branch chain, the $i$-th term develops the inverse spiral. The matrix obtains its bound spiral system as follows:

$$
\begin{aligned}
\$_1^r &= \begin{pmatrix} 0 & 1 & 0; & 0 & 0 & 0 \end{pmatrix} \\
\$_2^r &= \begin{pmatrix} 0 & 0 & 0; & 0 & 0 & 1 \end{pmatrix}
\end{aligned}
\tag{3}
$$

The constrained spiral system contains two constrained spirals. The $\$_1^r$ is the force constraint in the y-direction and $\$_2^r$ is a force couple constraint in the z-direction. The mechanism has four identical branch chains and $\$_1^r$ provides four force constraints in a plane parallel to the platform's plane. The mechanism has two redundant constraints, as shown by the fact that only two of the four force constraints in the same plane are linearly independent, and the four constraints restrict the movement of the moving platform in the x- and y-directions. The $\$_2^r$ restricts the rotation of the moving platform about the z-axis. Hence, the mechanism offers three DOFs for the x-, y-, and z-axis.

### 2.3. Inverse Solution Model

The position posture of the dynamic platform of the mechanism is known. The problem of solving the drive on the branch chain is done with the position inverse solution, as shown in Figure 1. The position vector $\mathbf{r} = \begin{bmatrix} x & y & z \end{bmatrix}^T$ at the center point O′ of the dynamic coordinate system O′-x′y′z′ located in the fixed coordinate system O-xyz is expressed as:

$$
\mathbf{r} = \mathbf{a}_i + q_i \mathbf{u}_i + l_i \boldsymbol{\omega}_i - \mathbf{b}_i, \ (i = 1, 2, 3, 4),
\tag{4}
$$

where $\mathbf{a}_i$ $\mathbf{b}_i$ are the measures of $A_i$ and $B_i$ in the fixed coordinate system O-xyz. The $q_i$ is the length of the moving sub, $\mathbf{u}_i$ is the unit vector of the moving sub, $l_i$ is the length of the fixed bar on the branch "$i$", and $\boldsymbol{\omega}_i$ is the unit vector of the fixed bar where $l_i$ is located.

$$\mathbf{a}_i = r_a \begin{pmatrix} \cos\alpha_i & \sin\alpha_i & 0 \end{pmatrix}^T, \tag{5}$$

$$\mathbf{b}_i = \mathbf{R}\mathbf{b}_{i0}, \tag{6}$$

$$\mathbf{b}_{i0} = r_b \begin{pmatrix} \cos\alpha_i & \sin\alpha_i & 0 \end{pmatrix}^T, \tag{7}$$

where $\alpha_i = (i-2)\frac{\pi}{2}$, $i = 1,2,3,4$ is the position angles of points $A_i$ and $B_i$ in the fixed and dynamic coordinate systems, respectively, and $r_a$ $r_b$ denote the radii of the static and dynamic platforms, respectively. Rewriting the closed-loop Equation (4) gives:

$$\mathbf{r} + \mathbf{b}_i - \mathbf{a}_i - l_i\boldsymbol{\omega}_i = q_i\mathbf{u}_i, \tag{8}$$

Taking the modulus of Equation (8) and squaring it produces:

$$\mathbf{r}^T\mathbf{r} - 2(\mathbf{a}_i + l_i\boldsymbol{\omega}_i - \mathbf{b}_i)^T\mathbf{r} + \mathbf{b}_i{}^T\mathbf{b}_i - 2(\mathbf{a}_i + l_i\boldsymbol{\omega}_i)^T\mathbf{b}_i + \mathbf{a}_i^T\mathbf{a}_i + 2l_i\mathbf{a}_i{}^T\boldsymbol{\omega}_i + l_i^2 - q_i^2 = 0, \ i = 1,2,3,4 \tag{9}$$

Solving Equation (9) according to the assembly pattern of the mechanism:

$$q_i = \sqrt{\mathbf{r}^T\mathbf{r} - 2(\mathbf{a}_i + l_i\boldsymbol{\omega}_i - \mathbf{b}_i)^T\mathbf{r} + \mathbf{b}_i{}^T\mathbf{b}_i - 2(\mathbf{a}_i + l_i\boldsymbol{\omega}_i)^T\mathbf{b}_i + \mathbf{a}_i^T\mathbf{a}_i + 2l_i\mathbf{a}_i{}^T\boldsymbol{\omega}_i + l_i^2}, \ i = 1,2,3,4 \tag{10}$$

$$\mathbf{u}_i = \frac{1}{q_i}(\mathbf{r} + \mathbf{b}_i - \mathbf{a}_i - l_i\boldsymbol{\omega}_i), i = 1,2,3, \tag{11}$$

### 2.4. Positive Solution Model

The positive positional solution is the positional variables of the moving platform with known inputs of the four-leg drive rod lengths. Due to the coupling between the four legs of the mechanism, the position of the endpoint of the mechanism's moving platform is decided by the inputs of the four-leg drive rods.

Multiplying both ends of Equation (8) by their respective transpositions gives:

$$\mathbf{r}^T\mathbf{r} - 2(\mathbf{a}_i + l_i\boldsymbol{\omega}_i - \mathbf{b}_i)^T\mathbf{r} + \mathbf{b}_i{}^T\mathbf{b}_i - 2(\mathbf{a}_i + l_i\boldsymbol{\omega}_i)^T\mathbf{b}_i + \mathbf{a}_i^T\mathbf{a}_i + 2l_i\mathbf{a}_i{}^T\boldsymbol{\omega}_i + l_i^2 - q_i^2 = 0, \ i = 1,2,3,4 \tag{12}$$

i.e.,

$$\mathbf{r}^T\mathbf{r} - 2(\mathbf{a}_1 + l_1\boldsymbol{\omega}_1 - \mathbf{b}_1)^T\mathbf{r} + \mathbf{b}_1{}^T\mathbf{b}_1 - 2(\mathbf{a}_1 + l_1\boldsymbol{\omega}_1)^T\mathbf{b}_1 + \mathbf{a}_1^T\mathbf{a}_1 + 2l_1\mathbf{a}_1{}^T\boldsymbol{\omega}_1 + l_1^2 - q_1^2 = 0 \tag{13}$$

$$\mathbf{r}^T\mathbf{r} - 2(\mathbf{a}_2 + l_2\boldsymbol{\omega}_2 - \mathbf{b}_2)^T\mathbf{r} + \mathbf{b}_2{}^T\mathbf{b}_2 - 2(\mathbf{a}_2 + l_2\boldsymbol{\omega}_2)^T\mathbf{b}_2 + \mathbf{a}_2^T\mathbf{a}_2 + 2l_2\mathbf{a}_2{}^T\boldsymbol{\omega}_2 + l_2^2 - q_2^2 = 0 \tag{14}$$

$$\mathbf{r}^T\mathbf{r} - 2(\mathbf{a}_3 + l_3\boldsymbol{\omega}_3 - \mathbf{b}_3)^T\mathbf{r} + \mathbf{b}_3{}^T\mathbf{b}_3 - 2(\mathbf{a}_3 + l_3\boldsymbol{\omega}_3)^T\mathbf{b}_3 + \mathbf{a}_3^T\mathbf{a}_3 + 2l_3\mathbf{a}_3{}^T\boldsymbol{\omega}_3 + l_3^2 - q_3^2 = 0 \tag{15}$$

$$\mathbf{r}^T\mathbf{r} - 2(\mathbf{a}_4 + l_4\boldsymbol{\omega}_4 - \mathbf{b}_4)^T\mathbf{r} + \mathbf{b}_4{}^T\mathbf{b}_4 - 2(\mathbf{a}_4 + l_4\boldsymbol{\omega}_4)^T\mathbf{b}_4 + \mathbf{a}_4^T\mathbf{a}_4 + 2l_4\mathbf{a}_4{}^T\boldsymbol{\omega}_4 + l_4^2 - q_4^2 = 0 \tag{16}$$

Subtracting Equation (14) from Equation (13) gives:

$$2(\mathbf{a}_1 - \mathbf{a}_2 + l_1\boldsymbol{\omega}_1 - l_2\boldsymbol{\omega}_2 + \mathbf{b}_2 - \mathbf{b}_1)^T\mathbf{r} = 2\left[(\mathbf{a}_2 + l_2\boldsymbol{\omega}_2)^T - (\mathbf{a}_1 + l_1\boldsymbol{\omega}_1)^T\right]$$
$$+ (\mathbf{a}_1^T\mathbf{a}_1 - \mathbf{a}_2^T\mathbf{a}_2) + \left(\mathbf{b}_1^T\mathbf{b}_1 - \mathbf{b}_2^T\mathbf{b}_2\right) + 2(l_1\mathbf{a}_1^T\boldsymbol{\omega}_1 - l_2\mathbf{a}_2^T\boldsymbol{\omega}_2) + l_1^2 - l_2^2 + q_2^2 - q_1^2 \tag{17}$$

Subtracting Equation (15) from Equation (13) results in the following:

$$2(\mathbf{a}_1 - \mathbf{a}_3 + l_1\boldsymbol{\omega}_1 - l_3\boldsymbol{\omega}_3 + \mathbf{b}_3 - \mathbf{b}_1)^T\mathbf{r} = 2\left[(\mathbf{a}_3 + l_3\boldsymbol{\omega}_3)^T - (\mathbf{a}_1 + l_1\boldsymbol{\omega}_1)^T\right]$$
$$+ (\mathbf{a}_1^T\mathbf{a}_1 - \mathbf{a}_3^T\mathbf{a}_3) + \left(\mathbf{b}_1^T\mathbf{b}_1 - \mathbf{b}_3^T\mathbf{b}_3\right) + 2(l_1\mathbf{a}_1^T\boldsymbol{\omega}_1 - l_3\mathbf{a}_3^T\boldsymbol{\omega}_3) + l_1^2 - l_3^2 + q_3^2 - q_1^2 \tag{18}$$

Rewriting Equations (17) and (18) into matrix form:

$$\begin{bmatrix} a_{11} & a_{12} & a_{13} \\ a_{21} & a_{22} & a_{23} \end{bmatrix} \begin{bmatrix} x \\ y \\ z \end{bmatrix} = \begin{bmatrix} g_1 \\ g_2 \end{bmatrix}, \tag{19}$$

$$a_{11} = 2(\mathbf{a}_{1x} - \mathbf{a}_{2x} + l_1\boldsymbol{\omega}_{1x} - l_2\boldsymbol{\omega}_{2x} + \mathbf{b}_{2x} - \mathbf{b}_{1x})$$
$$a_{12} = 2(\mathbf{a}_{1y} - \mathbf{a}_{2y} + l_1\boldsymbol{\omega}_{1y} - l_2\boldsymbol{\omega}_{2y} + \mathbf{b}_{2y} - \mathbf{b}_{1y})$$
$$a_{13} = 2(\mathbf{a}_{1z} - \mathbf{a}_{2z} + l_1\boldsymbol{\omega}_{1z} - l_2\boldsymbol{\omega}_{2z} + \mathbf{b}_{2z} - \mathbf{b}_{1z})$$
$$a_{21} = 2(\mathbf{a}_{1x} - \mathbf{a}_{3x} + l_1\boldsymbol{\omega}_{1x} - l_3\boldsymbol{\omega}_{3x} + \mathbf{b}_{3x} - \mathbf{b}_{1x})$$
$$a_{22} = 2(\mathbf{a}_{1y} - \mathbf{a}_{3y} + l_1\boldsymbol{\omega}_{1y} - l_3\boldsymbol{\omega}_{3y} + \mathbf{b}_{3y} - \mathbf{b}_{1y})$$
$$a_{23} = 2(\mathbf{a}_{1z} - \mathbf{a}_{3z} + l_1\boldsymbol{\omega}_{1z} - l_3\boldsymbol{\omega}_{3z} + \mathbf{b}_{3z} - \mathbf{b}_{1z})$$
$$g_1 = c_1 - c_2$$
$$g_2 = c_1 - c_3$$
$$c_i = \mathbf{b}_i{}^T\mathbf{b}_i - 2(\mathbf{a}_i + l_i\boldsymbol{\omega}_i)^T\mathbf{b}_i + \mathbf{a}_i^T\mathbf{a}_i + 2l_i\mathbf{a}_i{}^T\boldsymbol{\omega}_i + l_i^2 - q_i^2, i = 1, 2, 3, 4$$

Treating $z$ in Equation (19) as a known quantity:

$$x = f_1(z) = m_2 + n_2 z, \tag{20}$$

$$y = f_2(z) = m_1 + n_1 z, \tag{21}$$

$$m_1 = (a_{11}g_2 - a_{21}g_1)/(a_{12}a_{21} - a_{11}a_{22})$$
$$m_2 = (a_{22}g_1 - a_{12}g_2)/(a_{12}a_{21} - a_{11}a_{22})$$
$$n_1 = (a_{11}a_{23} - a_{13}a_{21})/(a_{12}a_{21} - a_{11}a_{22})$$
$$n_2 = (a_{22}a_{13} - a_{12}a_{23})/(a_{12}a_{21} - a_{11}a_{22})$$

Substituting Equations (20) and (21) into Equation (13):

$$kz^2 + ez + f = 0, \tag{22}$$

$$k = n_1^2 + n_2^2 + 1, \tag{23}$$

$$e = 2m_2 n_2 + 2m_1 n_1 - 2(\mathbf{a}_{1z} + l_1\boldsymbol{\omega}_{1z} - \mathbf{b}_{1z}) - 2(\mathbf{a}_{1x} + l_1\boldsymbol{\omega}_{1x} - \mathbf{b}_{1x})n_2 - 2(\mathbf{a}_{1y} + l_1\boldsymbol{\omega}_{1y} - \mathbf{b}_{1y})n_1 \tag{24}$$

$$f = m_2^2 + m_1^2 - 2(\mathbf{a}_{1x} + l_1\boldsymbol{\omega}_{1x} - \mathbf{b}_{1x})m_2 - 2(\mathbf{a}_{1y} + l_1\boldsymbol{\omega}_{1y} - \mathbf{b}_{1y})m_1 + c_1 \tag{25}$$

Thus, solving for $x$, $y$, and $z$ results in the following:

$$z = \tfrac{1}{2k}\left(-e - \sqrt{e^2 - 4kf}\right)$$
$$x = f_1(z), y = f_2(z)$$
$$\mathbf{r} = \begin{bmatrix} x & y & z \end{bmatrix}^T$$

## 3. Design of Dynamic Secondary Mirror Truss Adjustment Mechanism

Most large ground-based reflecting optical telescopes follow Cassegrain optical design. The structural design of large telescopes often uses a single or multi-layer truss structure as the main support structure. The main structure integrates with welded ring beams, and quadruple wing beam structures support the secondary mirror chamber, as shown in Figure 3 [15]. Suppose the secondary mirror truss is integrated with the SMAM. The secondary mirror adjusts the secondary mirror trusses. This mitigation reduces the overall height and mass of the telescope and increases the resonance frequency.

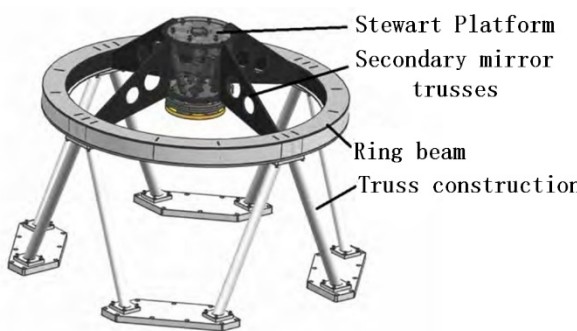

**Figure 3.** Secondary mirror support structure.

Based on the above kinematic modeling theory, a new dynamic secondary mirror truss adjustment mechanism is designed to replace the conventional Hexapod stage and adjust the secondary mirror's attitude by adjusting the secondary mirror truss. This study involves a dynamic secondary mirror truss adjustment mechanism, including a ring beam, a secondary mirror chamber, eight blades, and four correction units. The ring beam is fixed to the truss supporting the secondary mirror, and its components (the four alignment units) are fixed to the four sides of the ring beam. Each alignment unit is connected to the secondary mirror chamber by two blades. The chamber blades are applied with a preload so that the blades experience some tension regardless of the position of the secondary mirror chamber. The alignment unit consists of a stepper motor, worm gear reducer, eccentric cam, blade spring, moveable block, and absolute angle encoder. Figure 4 shows the overall structure of this SMAM.

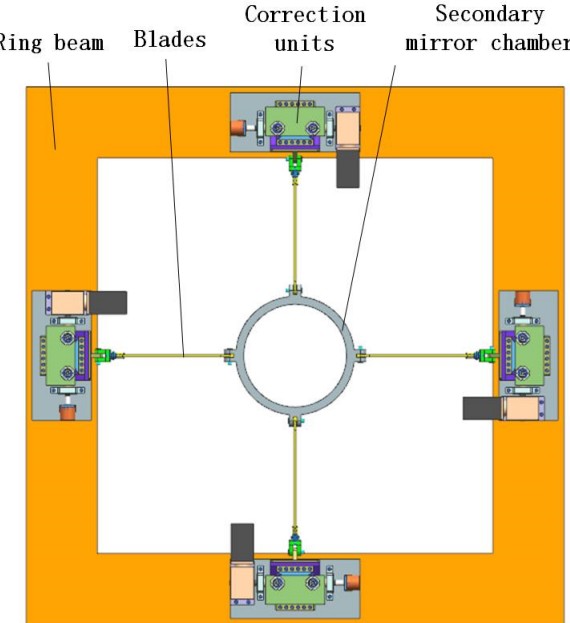

**Figure 4.** The overall structure of the SMAM.

The dynamic secondary mirror truss adjustment mechanism uses the movement of the four correction units along the optical axis to control the position of the secondary mirror. When the four correction units move in synchronization, the secondary mirror moves in the direction of the optical axis. When the four correction units are in different positions, a height difference is created between the two sides of the secondary mirror, thus controlling the deflection of the secondary mirror. The correction unit is a flat-bottomed direct-acting cam mechanism, as shown in Figure 5. The worm gear reducer reduces the speed and increases the torque. A stepped shaft and a flat key transmit the torque through an eccentric

cam, which acts on the removable block. By compressing the spring, the movable block is brought into tight contact with the eccentric cam, and the position of the movable block is controlled by adjusting the angle of rotation of the stepper motor to regulate the angle of rotation of the eccentric cam. The end of the shaft is coupled with an absolute angle encoder, which monitors and feeds back the position of the eccentric cam in real-time. The four removable blocks are adjusted separately so that the secondary mirror chamber can be translated (Tz) in the direction of the optical axis and deflected along the Rx and Ry of the vertical optical axis to achieve three DOFs. The two DOFs of movement along the x and y can be adjusted manually using pre-tightening bolts connecting the blades to the correction units.

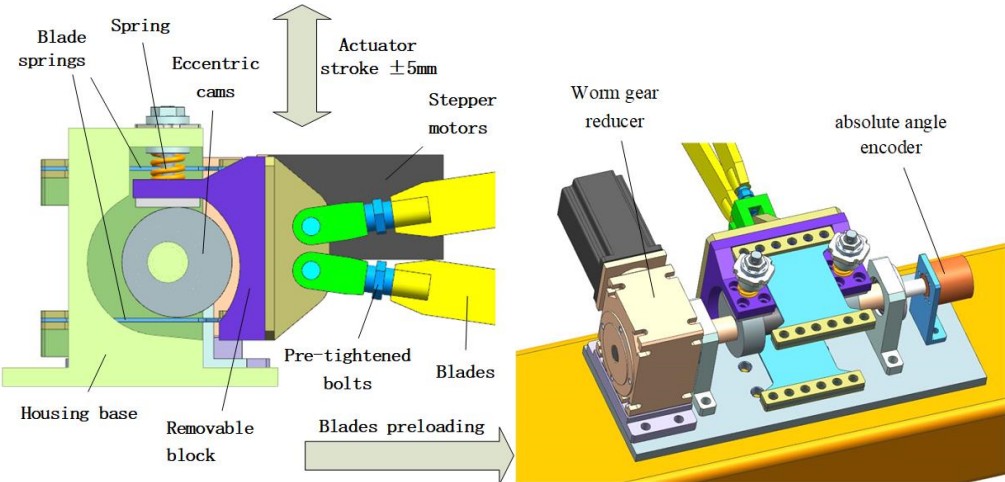

**Figure 5.** Correction unit mechanism.

## 4. Finite Element Simulation Analysis

### 4.1. Statistical Analysis

The lower base plate of the ring beam is restrained, and a preload force is applied to the blades to prevent flexural deformation due to pressure on the blades. A force of size 2000 N along the blade direction is added to the lower four blades, and a force of size 200 N along the blade direction is added to the upper four blades. These forces counteract the gravity of the secondary mirror assembly, as shown in Figure 6.

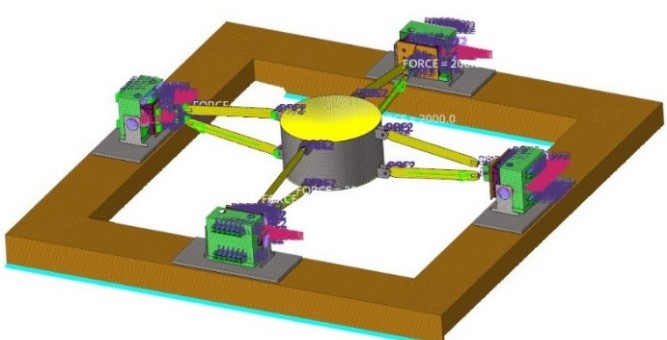

**Figure 6.** Finite element model of the SMAM.

When the telescope is at different zenith angles, a statistical analysis of the model yields the secondary mirror chamber deformation curve in different directions with the zenith angle, as shown in Figure 7.

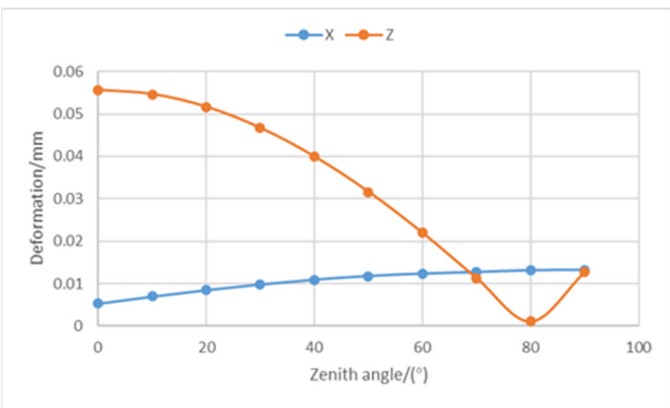

**Figure 7.** Deformation of the mirror chamber under gravity at different zenith angles.

The deformation in the x-direction increases with the zenith angle, and the deformation in the z-direction decreases first and then increases with the increase of the zenith angle. Since the zenith angle increases, the deformation of the secondary mirror chamber along the z-direction increases due to the different sizes of the preloaded blades.

Since the secondary mirror truss only introduces translation errors and does not include tilt errors, the effect of the x-directional shift on the imaging of the telescope system is calculated. For the Cassegrain system, the translation of the secondary mirror causes a tangential coma difference in the system, i.e.,

$$(\delta u)''_c = \frac{-3m_2}{16N^2}\left[\left(m_2^2 - 1\right) + \frac{1}{1 - R_A}\right]\frac{l}{f} \times 206265$$

where $m_2$ is the secondary mirror magnification, $N$ is the system focal ratio, $R_A$ is the axial light area obstruction ratio due to the secondary mirror, $f$ is the system focal length, and $l$ is the tangential offset of the secondary mirror. The obtained result is −0.0058, which satisfies the requirements for use.

*4.2. Modal Analysis*

Modal analysis is an effective method for deciding the vibration characteristics of a mechanism. The mechanism's inherent frequencies and vibration modes can be obtained for each order, which helps to avoid resonance due to insufficient stiffness in the structure's design and ensures the system's dynamic characteristics. Table 1 tabulates the first ten orders of modal values and vibration modes of the structure. The first-order mode value is approximately 61.24 Hz, and the maximum amplitude point in the secondary mirror chamber is due to the thin thickness of the blade spring. This tends to vibrate up and down. The second-order mode is the secondary mirror chamber rotation, and the maximum amplitude point is located on the blade. The mode value is approximately 74.77 Hz. The third- and fourth-order modes are the secondary mirror chamber translation modes. The two orders of modes occur due to the insufficient transverse stiffness of the correction unit. The eighth-order modes are the blade twisting vibration; these four orders are due to the insufficient blade stiffness, and the modal value is about 348.73 Hz. The ninth- and tenth-order modes are complex: the blade's bending vibration and the calibration unit's vibration. The modal value is about 434.64 Hz. Figure 8 shows the first two orders.

**Table 1.** First tenth-order modal values of the SMAM.

| Modal | 1 | 2 | 3~4 | 5~8 | 9~10 |
|---|---|---|---|---|---|
| Frequency/Hz | 61.24 | 74.77 | 157.33 | 348.73 | 434.64 |
| Vibration type | The vibration of the secondary mirror chambers up and down | Secondary mirror chamber rotation | Secondary mirror chamber translation mode | Blade twist vibration | Complex Modes |

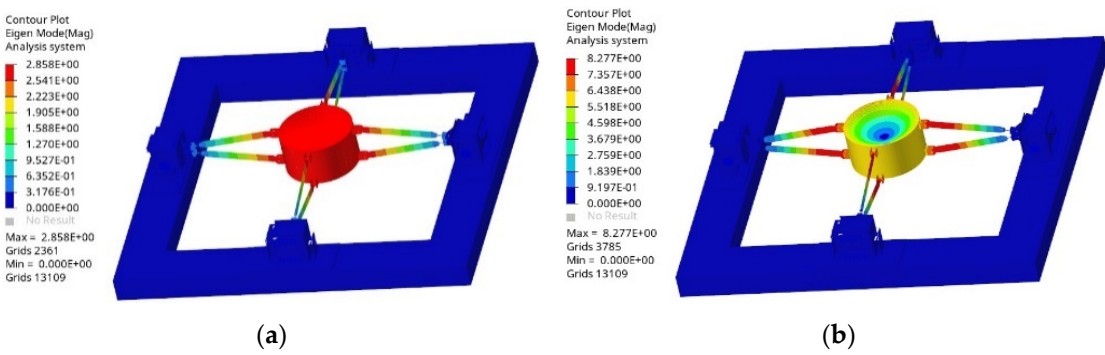

(**a**) (**b**)

**Figure 8.** (**a**) First-order modal vibration diagram; (**b**) Second-order mode vibration diagram.

## 5. Experimental Validations

Figure 9 shows the prototype of the SMAM. For the secondary mirror position adjustment, the actual motion performance of the prototype of the SMAM is tested [16,17], which includes the motion resolution test of the SMAM, the motion accuracy test, and the motion space test.

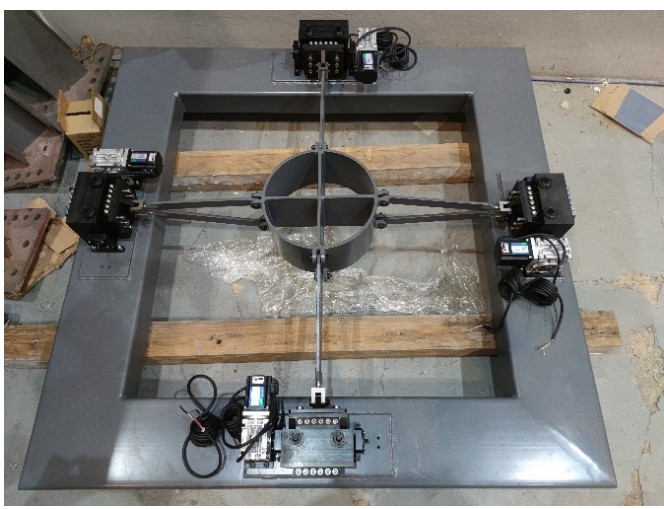

**Figure 9.** Prototype of the SMAM.

### 5.1. Motion Resolution Tests

The resolution of the movement of the adjustment mechanism was measured using a micrometer with an accuracy of one μm. The micrometer was placed at the center of the secondary mirror barrel to measure the displacement of the secondary mirror barrel in the z-direction, as shown in Figure 10. According to the design resolution, the adjustment mechanism's resolution was evaluated by starting with 1 μm and increasing the step size with 1 μm increments. When the step size reached around 5 μm, the output motion was more stable than the initial state; therefore, the resolution of the sub-mirror adjustment mechanism in the z-direction was evaluated in 5 μm steps. A control program along the

z-direction in 25 steps monitored the gradual movement of the adjustment mechanism. Each step followed the measured displacement output of the SMAM along the z-direction, as shown in Figure 11.

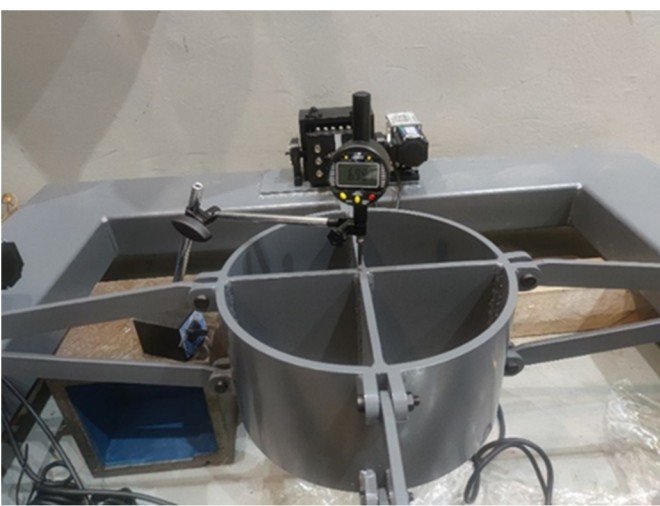

**Figure 10.** z-directional resolution test.

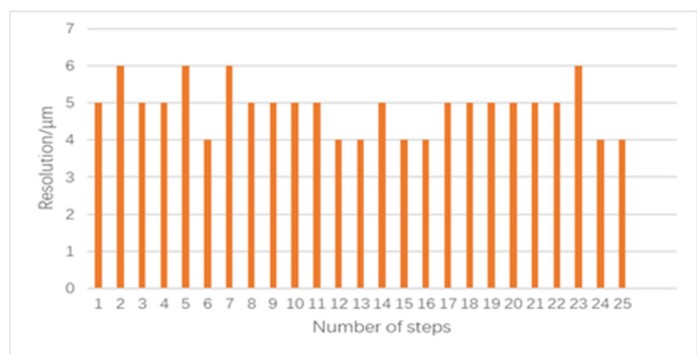

**Figure 11.** Results of motion resolution tests for displacement along the z-direction.

Figure 11 shows the results of the motion resolution test of the adjustment structure in the z-direction displacement, where the resolution of the adjustment mechanism in the z-direction translational motion is $5 \pm 1$ µm.

### 5.2. Motion Accuracy Tests

The SMAM measures the attitude accuracy after each movement step. Like the measurement of z-directional movement resolution, a micrometer is placed at the center of the secondary mirror barrel to measure the z-directional positional accuracy of the adjustment mechanism. Two micrometers are placed on each side of the secondary mirror barrel to calculate the deflection angle of the secondary mirror barrel. The height difference between the two sides of the secondary mirror barrel gives the positional accuracy of the adjustment mechanism, as shown in Figure 12. The test scheme for the position accuracy of the z-direction is as follows: Starting from –1 mm, drive the SMAM in steps of 0.1 mm within the range of –1 mm ~ +1 mm, then measure the position accuracy after achieving movement of each step. The specific measurement is taken from −1 mm to 0 mm, then +1 mm, and repeat the test five times. The specific measurement for attitude accuracy includes 0 to 400″, then 0 to +400″and finally to 0, and repeat the test five times.

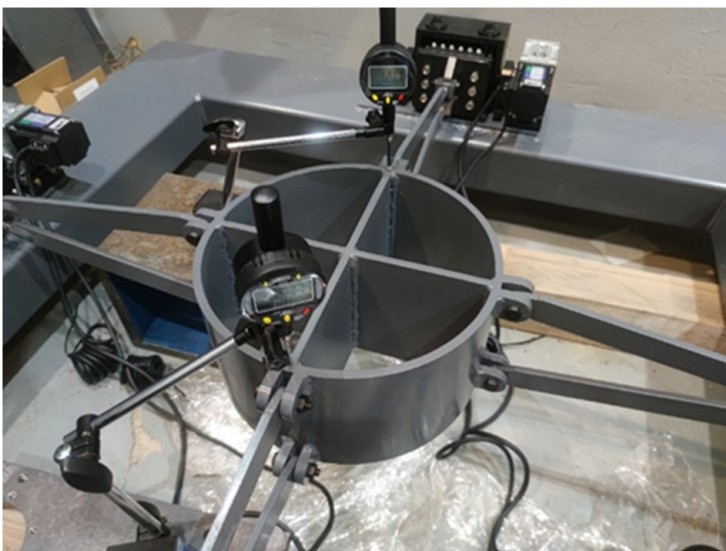

**Figure 12.** Deflection accuracy test with micrometers.

Figure 13 shows the results of the measurements in the z-direction. The number of drive steps is the horizontal coordinate, and the displacement measurement after each movement step is according to the vertical coordinates. The actual position of each measurement is shown as a green dot and the theoretical position with a red dot. The red dot of the theoretical position is placed above the green dot of the measurement result. The greener dots in the graph give the absolute error between the actual measured and theoretical positions. If the theoretical position coincides with the actual measurement position, the red dot at that position covers the green dot and is shown as a red dot.

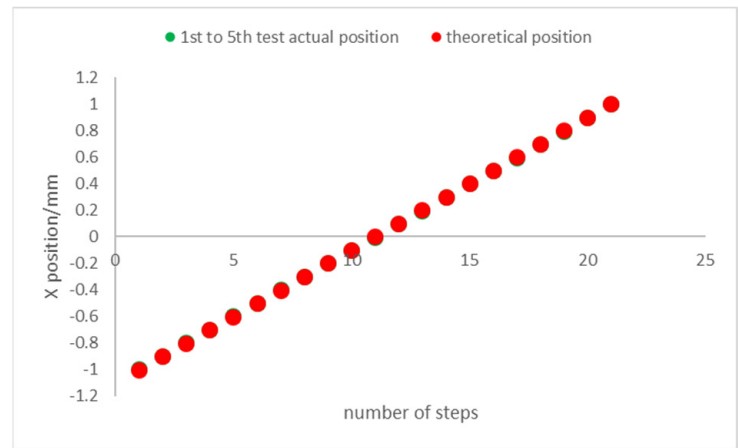

**Figure 13.** Measurement results of the z-directional position accuracy of the SMAM.

The absolute positioning error of the measurements after each step of the movement compared to the theoretical position is analyzed, as shown in Figure 14 below.

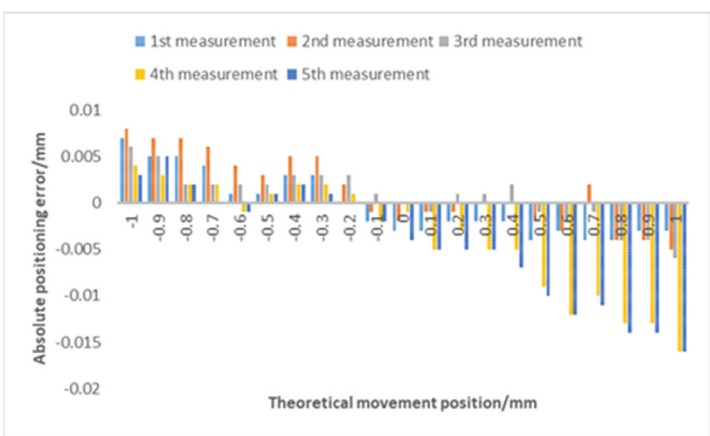

**Figure 14.** Absolute positioning error in the z-direction at each position point.

From Figure 14, the absolute positioning accuracy of the SMAM is better than ±0.016 mm in the z-direction. It starts from −1 mm and moves in steps of 0.1 mm within −1 mm ~ +1 mm. The absolute value of the repetitive positioning error at each position point can be obtained by processing the measurement results of the five repetitive movements, as shown in Figure 15.

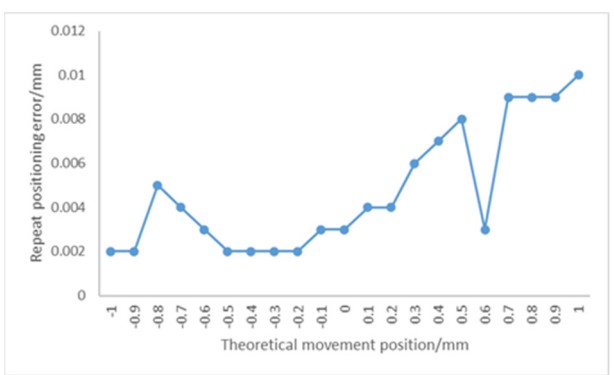

**Figure 15.** Repeat the positioning error in the z-direction at each position point.

From Figure 15, it is clear that the repeatable positioning accuracy of the SMAM is better than ±0.01 mm for translational movements along the z-direction, within the range of −1 mm to +1 mm.

The accuracy of the deflection attitude in the x-direction is measured according to the test protocol. Figure 16 shows the obtained results.

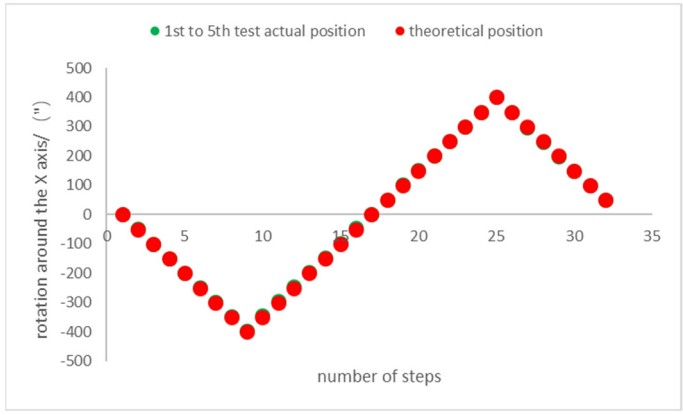

**Figure 16.** Attitude accuracy measurements for deflection around the x-direction.

Figure 17 shows the results of the absolute positioning error test of the deflection attitude in the x-direction of the SMAM.

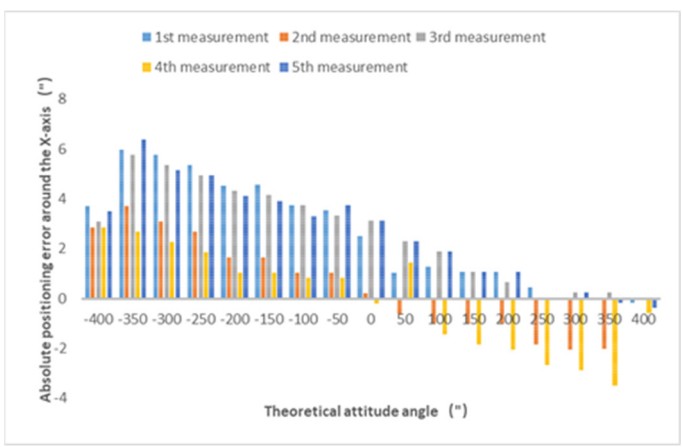

**Figure 17.** Absolute positioning error of the attitude deflected around the x-direction.

According to Figure 17, the absolute positioning accuracy of the attitude angle is better than 6.4″ when the deflection in the x-direction is in the range of −400″ to +400″. The measurement results of the five repeated movements are used to obtain the absolute error values of the repeated positioning of the deflection attitude angle in the x-direction, as shown in Figure 18.

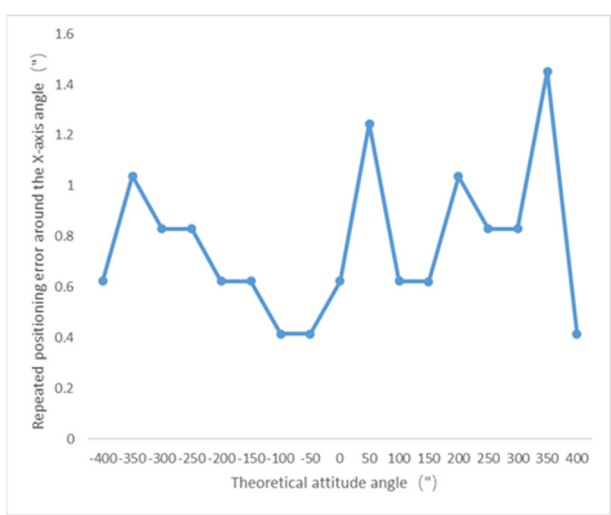

**Figure 18.** Repeat the positioning error of attitude angle at each position point.

The SMAM demonstrates an attitude angle repeatability accuracy better than ±1.5″ in the range of −400″ to +400″along the x-direction. The positional accuracy in other directions is also measured by the same method. After the accuracy test, the repeatable positioning accuracy of the SMAM is acceptable, and the difference between the absolute and repeatable positioning accuracy is found to be significant. When the stroke range and step length are more prominent, the SMAM's absolute positioning accuracy error also becomes larger. The analysis shows that this is caused by the assembly errors of the adjustment mechanism, the cam zeroing errors, and the errors during measurement. Therefore, it is necessary to analyze the absolute positioning accuracy of the SMAM.

*5.3. Practical Workspace Testing of the Mechanism*

The working space of the SMAM is the collection of all the spatial areas that the adjustable secondary mirror can reach, and the size of the working space is an essential

technical indicator of the SMAM. For analysis, the space of movement that can be reached by the center point of the secondary mirror chamber is evaluated. As the SMAM requires at least three DOFs to describe its attitude in any state, the working space of the SMAM can be divided into two parts: position space and attitude space. The position space is defined as the position that can be reached by the center point of the secondary mirror chamber throughout its range of travel. The attitude space is the maximum deflection attitude angle the secondary mirror chamber can reach in both the x- and y-directions once the chamber's center is known. According to the calibration requirements of the telescope's secondary mirror, the position and attitude of the secondary mirror must be corrected in several DOFs. Therefore, the position workspace and the SMAM's attitude workspace are measured separately.

The operating range of the secondary mirror adjustment is measured by controlling the movement of the SMAM prototype and the movement of the adjustment mechanism to the boundary position of the working space. By making several attempts in each direction, it is possible to determine whether the SMAM can move into place or move further. The results for the actual working space for the translation of the SMAM along the z-direction and the deflection movement around the x- and y-directions are given in Table 2.

**Table 2.** Results of the actual working space measurement of the SMAM.

| Direction of Movement | Stroke Measurement Results | Design Requirements |
|---|---|---|
| z-directional translation | $\pm5.19$ mm | $\pm5$ mm |
| x-directional deflection | $\pm0.577°$ | $\pm0.3°$ |
| y-directional deflection | $\pm0.574°$ | $\pm0.3°$ |

## 6. Results and Discussion

The dynamic secondary mirror truss adjustment mechanisms designed in this study provide a particular reference value for designing SMAMs for large aperture telescopes. As an essential part of the telescope system, the telescope's secondary mirror truss adjustment mechanism must be designed in a large mode to meet the control system requirements for control bandwidth. The designed SMAM has an absolute accuracy of better than 16 μm for movement in the z-direction and 20 μm for requirements and an absolute positioning accuracy of better than 6.4" for deflection in the x-direction and 8" for requirements. In the travel also, all meet the requirements of use, as shown in Table 2.

In the future, the study will consider other important factors, including the adjustment speed of the mechanism and the effect of different temperatures on the adjustment accuracy. The core components will be optimized to improve the resonance and increase the resolution and positioning accuracy.

**Author Contributions:** Conceptualization, B.L. and F.W.; methodology, B.L.; software, B.L.; validation, B.L., F.W. and B.Z.; formal analysis, B.L.; investigation, B.L.; resources, B.L.; data curation, B.L.; writing—original draft preparation, B.L.; writing—review and editing, B.L.; visualization, B.L.; supervision, B.L.; project administration, B.L.; funding acquisition, F.W. All authors have read and agreed to the published version of the manuscript.

**Funding:** This research was funded by The Jilin Science and Technology Development Program, grant number 20210402065GH; Excellent Member of Youth Innovation Promotion Association CAS, grant number Y202053; International Partnership Program of the Chinese Academy of Sciences, grant number 181722KYSB20200001; National Natural Science Foundation of China (NSFC); grant number 11973040; and National Key R&D Program of China; grant number 2017YFE0102900.

**Institutional Review Board Statement:** Not applicable.

**Informed Consent Statement:** Not applicable.

**Data Availability Statement:** No new data were created.

**Conflicts of Interest:** The authors declare no conflict of interest.

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
