# Peer review of "Design of a Dynamic Secondary Mirror Truss Adjustment Mechanism for Large Aperture Telescopes"

_applsci, doi:10.3390/app13021058_

Round 1

Reviewer 1 Report

My reply to Manuscript ApplSci-2117761-peer-review-v1
In the abstract, line 17-18, ‘’accuracy better than 6.4 arcseconds’’, it seems that this telescope is not relevant
for astronomical purposes (which generally require 1 arcsec image quality or better). Then, it could be
interesting for the reader to specify what this telescope could be used for, since you mention in the introduction
some references to telescopes – as VST, LSST and others – dedicated to astronomical studies. However, on L 47
and 48, is stated that this is for ‘’a large vehicle-mounted telescope’’; this could be also mentioned in the
abstract.

Figure 3 mentions a “Stewart Platform’’ thus implying a type of parallel interface made of triangular tripods with
six connection actuators on each face. Then from this figure the link between the ‘’Stewart Platform’’ and
secondary mirror is movable. Again, in Figure 3, the ‘’Truss construction’’ hexapod could be movable. Then,
there would be two such hexapod systems. For clarity, it would be convenient to develop the text associated to
Figure 3.

Figures 4, 5, and associated text, concerning the ‘’Secondary mirror truss[es?]’’ (in Fig.3) made of four arms or
height blades, is interesting but would gain to be clarified: From section 3, I recommend to make at least a
paragraph for each part relative to a figure.

Given numerous repetitions of ‘’secondary mirror adjustment mechanism’’ it would be useful, for clarity in the
reading, to create the abbreviation ‘SMAM’ starting at line 135 or before, after definition of this abbreviation.

Find some other comments to improve clarity:

- L 34, ‘’comet aberrations’’ is not appropriate and the sentence is unclear. It would be better to write: ‘tip/tilt,
which provides elimination of third-order coma aberrations caused...’

- L 68, ’B 1B2B3B4 ‘ should be better

- L 87, ’ith’ or ’i-th’ would be better

- L 103, after eq.(4), ’where’

- L 105, ’branch i

- L 106, after eq.(7), ’where’

- L 127, this sentence is better in italic: ’for x, y’. z is missing here; to be added ? Clarify this sentence for these
displacements

- L 165-166, ‘’control’’ is repeated many times

- L 169-170, ‘’moved (Tz)’’ hold for ‘translated Tz’ ? ‘’Rx” assumed a rotation around x ?

- L 184, ‘’hydrostatic’’: do you mean fluid mechanic’s ? This seems not appropriate

- L 196, ‘’comet’’ shall be called ‘coma’

- L 199, ‘’blocking ratio’’ assume ‘linear obstruction ratio’ or ‘area obstruction ratio’. Specify

- L 200, complete into ‘-0.00058 arcsec’. It would positive to create a line before this giving the values of all this
parameters

- L 242, ‘’Figure 10. Z-directional ...’’ could be completed by ‘View of the camera analyzer’ ?

- L 263, ‘’Figure 12. Deflection ...’’ could be completed by ‘with micrometers’

- L 325, ‘’degrees of freedom’’ should be replaced by ‘DOFs’

Reviewer 2 Report

Dear Authors,

I have finished reviewing your work. The results are very interesting and certainly relevant. The article can be published after making a minor revision. There are some comments (most of them are purely advisory in nature, but at the same time they could improve the quality and accessibility of the article for a wider readership).

General, but important comments:

1) The paper mentions large aperture telescopes. In my opinion it is necessary to quantify in the paper what apertures are mentioned to be large. 

2) It is crucial to mention for what types of telescopes and wavelength ranges the described adjustement mechanism is appliable and designed for. Are these only radio telescopes or it is suitable for optical, IR/FAR-IR etc.? Is this system universal for a wide operational wavelength range and telescope type? It is critical to provide an exact statement on it.

Following these two comments, in my opinion it is worth reflecting  necessary information not only in the text, but also in the abstract.

3) Although the article itself descirbes a large amount of work done, the last section #6 "Discussion" seems a little bit weak. I suggest to rename it to "Results and Discussion" and summarize the design and test work you've done in a form of highlights, providing an explicit conclusion why the obtained accuracies are acceptable and for what requirements.

4) It is hard to distinguish the dots on the Figures 13 and 16. Try to change the size of the dots of different color, or to change the scale, so it would be better to see the differences of the measured and calculated values.

Next, I provide some minor comments on the text (most refer to the language style/typos):

1) Page 1, line 31, "The secondary mirror" -> "It" (secondary mirror repeats)

2) Page 1, line 33, "The secondary mirror" -> "Adjustement mechanism" (secondary mirror repeats)

3) Page 2, line 47. I'm concercend about the term "vehicle-mounted telescope". It needs to be described in more detail. For now its not clear of what telescope type you're speaking about. For example, you may check Miyoshi et al, Advances in Astronomy, 2016. They describe a vehicle-mounted submm telescope system. There are also concepts of such optical and radio telescopes as well.

4) Page 2, line 47. Remove duplicate sentence "This paper deals with the design..."

5) Page 2, line 59. "The mechanism sketch..." -> "The sketch"

6) Page 3, line 76. "Where c and s..." -> "Where "c" and "s"..."

7) Page 3, line 82. "chain i system" -> "chain "i" system"

8) Page 3, line 87. "ith term" -> "i-th term"

9) Page 4, line 105. "on brach i" -> "on the brach "i""

10) Page 4, line 114. "the positional position" -> "the position" (I guess it is suitable to leave just as "the position" to avoid tautology)

11) Page 7, line 165 and 166. "is conrtolled by controlling ... moto to control". Please revise the sentence to avoid multiple repeats of the same word.

12) Page 9, line 194. "effect of the x-directional" -> "effect of the X-directional"

13) Page 9, line 205. "which helps avoid" -> "which helps to avoid"

14) Page 9, line 206. "Table 1 shows the structure's..." -> "Table 1 shows the first ten orders of modal values and vibration modes of the structure"

15) Page 10, line 238. "25 steps controlled the" -> "25 steps monitored the"

16) Page 12, line 275. "measurements of the position after" -> "measurements after" (it is clear here that you speak about the position measurements)

17) Page 13, line 290. Please remove "of the secondary mirror adjustement mechanism". It is clear that the sentence refers to the adjustement system

18) Page 13, line 291-292. Please provide some comments on the results shown on the Figure 16. Do they meet the requirements, etc.?

19) Page 13, line 296. "X direction" -> "X-direction"

20) Page 13, line 296. Please remove "of the secondary mirror adjustement mechanism". It is clear that the sentence refers to the adjustement system

21) Page 13, line 300. "X direction" -> "X-direction"

22) Page 13, line 300. Please remove "of the secondary mirror adjustement mechanism". It is clear that the sentence refers to the adjustement system

23) Page 13, line 303. "X direction" -> "X-direction"

24) Page 13, line 303. Please remove "of the secondary mirror adjustement mechanism". It is clear that the sentence refers to the adjustement system

25) Page 14, line 308. "X direction" -> "X-direction"

26) Page 14, line 311. "mirror adjustement mechanism is acceptable". Please specify acceptable for what? What requirements the accuracy meets?

27) Page 14, line 327. "parts position space, and attitude space" -> "parts: position space and attitude space"

28) Page 14, line 330. "X and Y directions" -> "X- and Y-directions"

29) Page 14, line 341. "Z direction" -> "Z-direction"

30) Page 14, line 341. "X and Y directions" -> "X- and Y-directions"

31) Page 15, line 349. "support adjustment" -> "truss adjustement"

32) Page 15, line 355. "the adjustement speed of the adjustement mechanism" -> "the adjustement speed of the mechanism"

Advisory comments:

1) The paper presents in sufficient detail the calculations describing the kinematic model. There is no doubt that the authors carefully approached those calculations. However, it may be worth shortening this part for easier perception. It is just a recommendation.

2) For a better visual demonstration, I would recommend that the authors provide a scheme showing where and how the system for the secondary mirror is mounted. Additionally, it could be useful to show the directions along which the secondary mirror (for example, using arrows) is adjusted in one of the figures that are already presented in the paper (for example, Fig. 4).

I would like to thank the Authors for their careful work and wish them success in their future acitivities.

With my best and kindest regards,

Reviewer.
